# Acute Kidney Injury after Endovascular Treatment in Patients with Acute Ischemic Stroke

**DOI:** 10.3390/jcm9051471

**Published:** 2020-05-14

**Authors:** Joonsang Yoo, Jeong-Ho Hong, Seong-Joon Lee, Yong-Won Kim, Ji Man Hong, Chang-Hyun Kim, Jin Wook Choi, Dong-Hun Kang, Yong-Sun Kim, Yang-Ha Hwang, Jin Soo Lee, Sung-Il Sohn

**Affiliations:** 1Department of Neurology, Keimyung University School of Medicine, Daegu 42601, Korea; quarksea@gmail.com (J.Y.); neurohong79@gmail.com (J.-H.H.); 2Department of Neurology, National Health Insurance Service Ilsan Hospital, Goyang 10444, Korea; 3Department of Neurology, Ajou University School of Medicine, Suwon 16500, Korea; editisan@hanmail.net (S.-J.L.); dacda@hanmail.net (J.M.H.); jinsoo22@gmail.com (J.S.L.); 4Department of Neurology, School of Medicine, Kyungpook National University, Daegu 41944, Korea; yw.kim23@gmail.com (Y.-W.K.); yangha.hwang@gmail.com (Y.-H.H.); 5Department of Neurosurgery, Keimyung University School of Medicine, Daegu 42601, Korea; ppori2k@naver.com; 6Department of Radiology, Ajou University School of Medicine, Suwon 16500, Korea; radjwchoi@gmail.com; 7Department of Neurosurgery, School of Medicine, Kyungpook National University, Daegu 41944, Korea; kdhdock@hotmail.com; 8Department of Radiology, School of Medicine, Kyungpook National University, Daegu 41944, Korea; yongkim@knu.ac.kr

**Keywords:** ischemic stroke, acute kidney injury, contrast media, endovascular treatment, outcome

## Abstract

Acute kidney injury (AKI) is often associated with the use of contrast agents. We evaluated the frequency of AKI, factors associated with AKI after endovascular treatment (EVT), and associations with AKI and clinical outcomes. We retrospectively analyzed consecutively enrolled patients with acute ischemic stroke who underwent EVT at three stroke centers in Korea. We compared the characteristics of patients with and without AKI and independent factors associated with AKI after EVT. We also investigated the effects of AKI on functional outcomes and mortality at 3 months. Of the 601 patients analyzed, 59 patients (9.8%) developed AKI and five patients (0.8%) started renal replacement therapy after EVT. In the multivariate analysis, diabetes mellitus (odds ratio (OR), 2.341; 95% CI, 1.283–4.269; *p* = 0.005), the contrast agent dose (OR, 1.107 per 10 mL; 95% CI, 1.032–1.187; *p* = 0.004), and unsuccessful reperfusion (OR, 1.909; 95% CI, 1.019–3.520; *p* = 0.040) were independently associated with AKI. The presence of AKI was associated with a poor functional outcome (OR, 5.145; 95% CI, 2.177–13.850; *p* < 0.001) and mortality (OR, 8.164; 95% CI, 4.046–16.709; *p* < 0.001) at 3 months. AKI may also affect the outcomes of ischemic stroke patients undergoing EVT. When implementing EVT, practitioners should be aware of these risk factors.

## 1. Introduction

Acute kidney injury (AKI) is an acute worsening of renal function often associated with the use of contrast agents [1,2]. Although the understanding of contrast-associated AKI (CA-AKI) has improved, CA-AKI remains an important issue in procedures using contrast agents such as computed tomography angiography (CTA), computed tomography perfusion (CTP), and endovascular treatment (EVT). The risk of AKI is also increased by acute ischemic stroke itself [3]. AKI is not uncommon and may lead to poorer outcomes in ischemic stroke patients [4,5,6].

Recently, EVT has been established as a treatment for acute ischemic stroke [7,8,9]. Several studies of AKI in acute ischemic stroke patients have been performed, but research is still lacking on AKI in stroke patients who have undergone EVT. Prior to the popularity of EVT, there were pioneering AKI studies of patients who underwent EVT, but it was difficult to determine the characteristics of the AKI patients or the risk factors associated with AKI because of the relatively small number of included patients [10,11]. There has been recent research on the effect of baseline renal impairment on CA-AKI, but little is known about the factors associated with CA-AKI [12]. Therefore, in this study, we investigated the incidence of AKI, the risk factors associated with AKI after EVT, and the effect of AKI on the outcomes of ischemic stroke patients who underwent EVT.

## 2. Materials and Methods

### 2.1. Study Participants

This study is a retrospective analysis of data from the Acute Stroke due to Intracranial Atherosclerotic occlusion and Neurointervention Korean Retrospective (ASIAN KR) registry. Details of the registry have been previously published [13,14]. Briefly, data in the registry were collected from the patients of three university stroke centers in Korea from January 2011 to February 2016. During the study period, acute stroke patients who underwent EVT were consecutively enrolled. In this study, we included patients with an onset-to-puncture time of ≤24 h. All patient data were anonymized, and each patient was assigned an identification number. The data collection protocol was approved by the institutional review board of each hospital. 

### 2.2. Clinical Assessment and EVT Process

Imaging and clinical analyses were performed in a core lab after the de-identification process. The initial stroke severity and serum creatinine level were assessed at the time of arrival, before images were taken. Follow-up serum creatinine results were collected from the next day to 7 days after the baseline images. Stroke severity was evaluated using the National Institutes of Health Stroke Scale (NIHSS) score. Renal function was assessed using the estimated glomerular filtration rate (eGFR) with the Modification of Diet in Renal Disease formula. Baseline renal function was classified into four groups by eGFR values of 90 mL/min/1.73 m^2^ or above (Stage I), 60 to 89 mL/min/1.73 m^2^ (Stage II), 30 to 59 mL/min/1.73 m^2^ (Stage III), and under 30 mL/min/1.73 m^2^ (Stage IV and V). The number of patients with Stage V CKD was only three; therefore, they were classified together with Stage IV CKD. AKI was evaluated using the Kidney Disease Improving Global Outcomes criteria [15,16]. Patients were considered to have AKI if they had an increment in serum creatinine of 0.3 mg/dL within 48 h or an increment in serum creatinine 1.5 times that recorded at baseline within 7 days. An increment in serum creatinine 2.0–2.9 times that recorded at baseline within 7 days indicated AKI Stage 2. An increment in serum creatinine three or more times that recorded at baseline, an increase in serum creatinine to 4.0 mg/dL or more, or the initiation of renal replacement therapy indicated AKI Stage 3. We also investigated whether CTA was performed before EVT. The device used for the EVT procedure was chosen by the treating physician. A direct aspiration device or a stent retriever was recommended as a primary reperfusion device. Balloon guide catheters, intracranial or extracranial angioplasty, and/or stenting were implemented as needed. Either Visipaque (Iodixanol, GE healthcare, Marlborough, MA, USA) or Pamiray (Iopamidol, Dongkook Pharm., Seoul, Korea) was used as the contrast medium during the EVT procedure. The contrast dose was based on the prescribed records. However, it was recalculated based on cerebral angiography imaging and procedure because a large amount of the contrast was discarded during the procedure. First, the usual dose of contrast medium used in each hospital was checked according to the artery and procedure. The total amount of contrast agent administrated to the patients was retrospectively calculated by assigning each image series and routine procedural dose. Reperfusion status was evaluated using the modified thrombolysis in cerebral infarction (mTICI) grade on the final angiogram [17]. Successful reperfusion was defined as an mTICI grade of 2b or 3. Time intervals, including onset-to-door time, door-to-puncture time, and total procedure time, were assessed. The procedure time was defined as the time from puncture to the final angiogram. Hemorrhagic transformation was evaluated using follow-up computed tomography or magnetic resonance imaging. Intracerebral hemorrhages were classified in accordance with the European Cooperative Acute Stroke Study criteria [18]. Functional status was assessed using the modified Rankin scale (mRS) score. A poor clinical outcome was defined as an mRS of 3 or more at 3 months. If a patient’s preclinical mRS was 3 or more and the 3-month mRS did not worsen, the patient was not classified as having a poor clinical outcome. We also identified mortality at 3 months.

### 2.3. Statistical Analyses

Data are expressed as means ± standard deviations, medians (interquartile ranges (IQR)), or numbers (percentages), as statistically appropriate. We compared the imaging and clinical variables between the groups with and without AKI using chi-squared tests, independent Student’s *t*-tests, or Wilcoxon rank-sum tests, respectively. To identify the factors associated with AKI, we performed multivariate analysis after adjusting for age, sex, the initial stroke severity, baseline renal function, the performance of CTA before EVT, and factors with *p* < 0.1 in the univariate analysis. We also assessed the factors associated with a poor clinical outcome and mortality at 3 months. To investigate these associations, we performed a multivariate analysis after adjusting for the presence or stage of AKI, age, sex, the initial NIHSS score, baseline renal function, and factors with *p*-value < 0.1 in the univariate analysis. All *p*-values were two-tailed, and variables were considered significant at *p*-value < 0.05. All statistical analyses were performed using R version 3.6.2 (http://www.R-project.org).

## 3. Results

During the study period, a total of 720 patients were enrolled in the ASIAN KR registry. After excluding 21 patients because they had undergone EVT more than 24 h after onset, because of early mortality due to malignant brain edema, or because of hemodialysis before admission, 699 patients were eligible for this study. Of these, 98 patients lacked baseline or follow-up renal function tests. Finally, 601 patients (86.0%) were included in our analysis (Figure 1). The mean age of the included patients was 68.0 ± 12.2 years, and 333 patients (55.4%) were men. CTA before EVT was performed in 510 patients (84.9%), and the mean dose for CTA was 82.6 ± 7.6 mL. The mean dose of the contrast agent was 71.2 ± 37.2 mL, and 452 of the included patients (75.2%) had successful reperfusion. Patients with successful reperfusion used smaller amounts of the contrast agent than those with unsuccessful reperfusion (68.5 ± 36.0 vs. 79.1 ± 39.5 mL, *p* = 0.004). Most of the excluded patients (95 patients, 96.9%) were excluded due to missing follow-up creatinine levels. The excluded patients tended to be younger than the included patients (68.0 ± 12.2 years vs. 63.8 ± 12.6 years, *p* = 0.003) and had better baseline renal function (eGFR: 64.6 ± 26.1 vs. 72.4 ± 24.6 mL/min/1.73 m^2^, *p* = 0.005) and less severe stroke (initial NIHSS: 17 (13–21) vs. 14 (9–18), *p* < 0.001). None of the excluded patients required renal replacement therapy during their hospital stay. The excluded patients also showed better clinical outcomes at 3 months (mRS 3 (1–5) vs. 1 (0–2), *p* < 0.001) (Appendix A).

### 3.1. Factors Associated with Acute Kidney Injury

Among the 601 included patients, 59 (9.8%) developed AKI within 7 days of EVT. Of these, 22 (3.7%) were classified with AKI Stage 1, 14 (2.3%) were classified with AKI Stage 2, and 23 (3.8%) were classified with AKI Stage 3. Of the patients with AKI, renal replacement therapy was initiated in five patients (0.8%). The age and sex were similar between patients with and without AKI (Table 1). The amount of contrast medium used was higher in patients with AKI (69.1 ± 36.0 vs. 89.8 ± 42.9 mL, *p* = 0.001). In the multivariate analysis, diabetes mellitus (odds ratio, 2.341; 95% CI, 1.283–4.269; *p* = 0.005), contrast dose (odds ratio, 1.107 per 10 mL; 95% CI, 1.032–1.187; *p* = 0.004), and unsuccessful reperfusion (odds ratio, 1.909; 95% CI, 1.019–3.520; *p* = 0.040) were independently associated with the presence of AKI (Table 2). The performance of CTA before EVT was associated with an increased AKI risk, but the increase was not statistically significant (odds ratio, 2.112; 95% CI, 0.786–7.406; *p* = 0.181).

### 3.2. Factors Associated with Functional Outcome and Mortality at 3 Months

Follow-up was conducted at 3 months for all patients to determine functional outcome and mortality. There were 330 patients (54.9%) who showed poor functional outcomes at 3 months. Of the 59 patients with AKI, 52 (88.1%) showed poor outcomes, and only seven (11.9%) showed good outcomes (*p* < 0.001) (Appendix A). In the multivariate analysis, the presence of AKI was independently associated with a poor outcome (odds ratio, 5.145; 95% CI, 2.177–13.850; *p* < 0.001) (Table 3, Model 1). AKI Stage 2 (odds ratio, 13.709; 95% CI, 2.108–280.187; *p* = 0.022) and Stage 3 (odds ratio, 6.028; 95% CI, 1.452–42.593; *p* = 0.030) was also associated with a poor functional outcome at 3 months (Table 3, Model 2 and Figure 2). Baseline renal function showed an association with functional outcome (*p* value for trend = 0.005); however, it did not show an independent association in the multivariate analysis. The dose of contrast medium used was also associated with a poor functional outcome (odds ratio, 1.080 per 10 mL increase; 95% CI, 1.013–1.155; *p* = 0.021).

During the 3-month follow-up, 86 patients (14.3%) died. Of these, 29 (33.7%) had AKI. Of the 515 survivors, 30 (5.8%) had AKI (*p* < 0.001) (Appendix A). In the multivariate analysis, the presence of AKI was significantly associated with mortality (odds ratio, 8.164; 95% CI, 4.046–16.709; *p* < 0.001) (Table 4, Model 1). AKI Stage 2 (odds ratio, 20.845; 95% CI, 5.907–82.054; *p* < 0.001) and Stage 3 (odds ratio, 13.670; 95% CI, 4.740–41.925; *p* < 0.001) were also independently associated with mortality at 3 months (Table 4, Model 2). 

## 4. Discussion

This study assessed the frequency of AKI and the association between AKI and clinical outcomes in ischemic stroke patients who had undergone EVT. Our data showed that about 9.8% of the ischemic stroke patients developed AKI after EVT. However, renal replacement therapy was required for less than 1% of the patients who underwent EVT. The rate does not differ significantly from the 9.6% incidence rate of AKI after stroke in meta-analysis [5]. However, the incidence of AKI in our study is higher than those in CTA studies of ischemic stroke patients [19] or in studies of general ischemic stroke patients [4]. Our study included patients who underwent EVT and who had experienced a relatively severe stroke, which may have influenced the development of higher incidences of AKI. The rate in our study is also higher than in other recently reported studies using EVT patients. This is probably because the patients enrolled in this study are older, with worse ASPECTS and a lower successful reperfusion rate [12]. However, the result is lower than the 20.9% of stroke patients who were admitted to the neurology intensive care unit [6].

In this study, the presence of diabetes mellitus was independently associated with AKI after EVT in ischemic stroke patients. In a previous CTA study of ischemic stroke patients, diabetes mellitus was also associated with contrast-induced nephropathy [20]. Diabetes mellitus was also found to affect the occurrence of AKI in a previous study of patients who underwent percutaneous coronary interventions [21]. However, the precise pathophysiologic mechanisms of CA-AKI remain unclear [2]. Several studies suggested that AKI can be caused by ischemia due to the vasoconstrictive properties of the contrast media or the direct toxic effects of the contrast media on endothelial cells and renal tubules [22]. Patients with diabetes have an increased sensitivity to renal vasoconstrictors and renal ischemia, as well as decreased nitric-oxide-dependent vasodilatation [23,24]. Therefore, patients with diabetic nephropathy are more vulnerable to renal ischemia caused by a contrast medium.

The total amount of contrast medium and the final reperfusion status were also associated with AKI after EVT. Previous studies of patients undergoing coronary interventions have also identified these dose-dependent impairments of renal function [25,26]. In another study of acute myocardial infarction, the use of more than 100 mL of contrast agent increased the risk of AKI [27]. In coronary angiography, contrast medium refluxes into the aorta and renal arteries. However, contrast material administered into the intracranial arterial circulation is intravenous from the perspective of the kidneys. For this reason, it is somewhat different from coronary angiography. However, as our findings showed the relationship between the contrast dose and AKI, practitioners should keep in mind it is best to avoid the use of excessive amounts of contrast medium during EVT if possible.

Reperfusion status is directly related to the short-term prognosis as well as the long-term outcome in stroke patients with large artery occlusion. Patients with successful reperfusion required the use of fewer EVT techniques and smaller amounts of the contrast medium. In patients with poor reperfusion, the possibility of complications such as brain edema or pneumonia increases [28,29]. Using osmotic diuretics such as mannitol or broad-spectrum antibiotics may increase the risk of AKI [30,31]. It is well known that initial stroke severity is related to the risk of AKI [32,33]. At a given severity of stroke, the risk of AKI was higher in patients without successful reperfusion than in patients with successful reperfusion. This may be related to the fact that, unlike in other studies that did not consider reperfusion, the initial NIHSS score was not significantly associated with AKI risk in our study.

The performance of CTA before EVT was not independently associated with AKI after EVT. In previous CTA or CTP studies, intravenous contrast agents did not increase AKI risk in patients with acute stroke [34,35,36]. In a study of 12,508 propensity-score-matched patients, intravenous contrast material used for computed tomography did not increase AKI risk [37]. In a reperfusion study examining intravenous tissue plasminogen activator in acute ischemic stroke patients, CTA also did not affect renal function [38]. It is substantially higher following catheter-based procedures with intra-arterial contrast agent administration than it is following imaging procedures with intravenous contrast agent administration [1,39,40]. Although there is a tendency for this to raise the risk of AKI, the implementation of CTA is considered relatively safe in ischemic stroke patients, regardless of whether they undergo EVT.

Our study also showed that the presence of AKI and the severity of AKI were associated with poor functional outcomes and mortality at 3 months. It is well known that AKI affects mortality and functional outcomes in acute ischemic stroke patients [4,5]. CA-AKI also increases mortality, lengthens hospitalization, and increases the cost burden [41]. Therefore, interventions to prevent AKI may improve outcomes after stroke, especially in patients with diabetes mellitus or severe renal impairment [22].

There were several limitations to this study. First, it was a retrospective analysis of a prospectively enrolled registry, which may have led to bias. Second, as with other studies examining CA-AKI that occur after EVT, there is no control; therefore, it is not known whether AKI was caused by other factors. Third, about 14% of patients were excluded due to lack of a follow-up renal function test. These individuals had better baseline renal function prior to the EVT and showed better clinical outcomes. They probably had relatively good renal function in the admission period, which indicates AKI was likely overestimated. Fourth, we could not obtain an accurate amount of the contrast agent that is related to risk of AKI. Further study is needed to overcome these shortcomings and to confirm our results. However, our study is meaningful in that it investigated the factors related to the risk of AKI and the characteristics of patients with AKI after EVT. Moreover, the number of included patients was larger than that in previous studies.

## 5. Conclusions

In conclusion, the incidence of AKI after EVT was approximately 9.8%. Diabetes mellitus, the total amount of contrast medium, and unsuccessful reperfusion were independently associated with the development of AKI in patients who underwent EVT. AKI may also affect the outcomes of ischemic stroke patients undergoing EVT. When implementing EVT, practitioners should be aware of these risk factors.

## Figures and Tables

**Figure 1 jcm-09-01471-f001:**
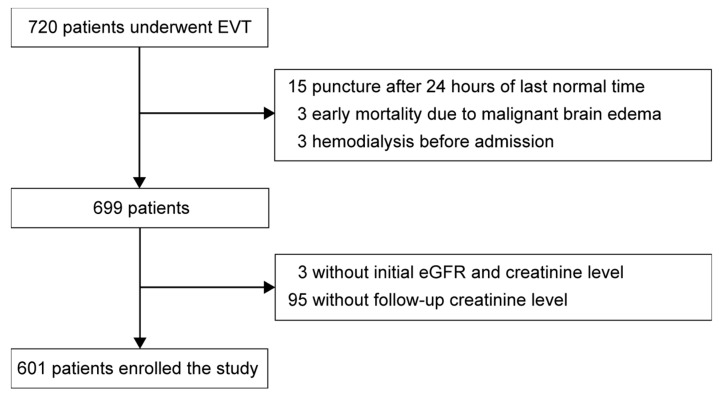
Patients enrollment. EVT, endovascular treatment; eGFR, estimated glomerular filtration rate.

**Figure 2 jcm-09-01471-f002:**
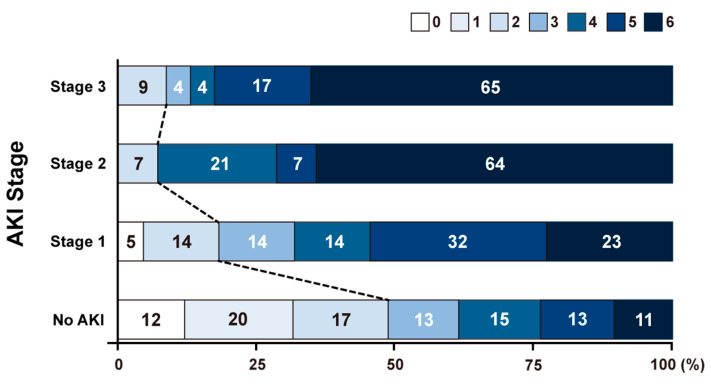
Modified Rankin Scale at 3 months according to acute kidney injury stage. AKI, acute kidney injury.

**Table 1 jcm-09-01471-t001:** Patient characteristics according to the presence of acute kidney injury.

	No Acute Kidney Injury (*n* = 542)	Acute Kidney Injury (*n* = 59)	*p*-Value
Age, years	67.9 ± 12.2	68.3 ± 11.8	0.827
Sex, men	307 (56.6)	26 (44.1)	0.088
Risk factors			
Hypertension	336 (62.0)	45 (76.3)	0.043
Diabetes mellitus	139 (25.6)	28 (47.5)	0.001
Atrial fibrillation	267 (49.3)	31 (52.5)	0.733
Dyslipidemia	160 (29.5)	20 (33.9)	0.584
Smoking	118 (21.8)	10 (16.9)	0.489
Previous stroke or history of TIA	96 (17.7)	11 (18.6)	>0.999
Medication prior to admission			
Antiplatelets	145 (26.8)	17 (28.8)	0.854
Anticoagulants	73 (13.5)	6 (10.2)	0.611
Statins	26 (4.8)	8 (13.6)	0.013
Baseline renal function			0.996 *
eGFR ≥90 mL/min/1.73 m^2^	84 (15.5)	14 (23.7)	
eGFR 60–89 mL/min/1.73 m^2^	194 (35.8)	17 (28.8)	
eGFR 30–59 mL/min/1.73 m^2^	233 (43.0)	19 (32.2)	
eGFR <30 mL/min/1.73 m^2^	31 (5.7)	9 (15.3)	
CTA before EVT	455 (83.9)	55 (93.2)	0.090
Contrast agent			0.807
Iodixanol	391 (72.1)	44 (74.6)	
Iopamidol	151 (27.9)	15 (25.4)	
Contrast dose, mL	69.1 ± 36.0	89.8 ± 42.9	0.001
Laboratory findings			
Hemoglobin, g/dL	13.5 ± 1.8	12.8 ± 2.1	0.025
White blood cells, ×10^9^/L	8.7 ± 3.4	9.8 ± 4.2	0.063
Platelets, ×10^9^/L	221 ± 69	226 ± 75	0.645
Glucose, mmol/L	7.7 ± 3.0	8.9 ± 3.8	0.029
Stroke-related factors			
NIHSS score on admission	17 (13–21)	19 (14.5–21.5)	0.022
ASPECTS †	7 (5–9)	5 (3–8)	0.004
Intravenous tPA	279 (51.5)	31 (52.5)	0.985
Onset to puncture time, min	270 (180–445)	251 (189–402)	0.614
Procedure time, min	61.5 (43–90)	65 (40–126)	0.378
Unsuccessful reperfusion (mTICI 2a or less)	126 (23.2)	23 (39.0)	0.012
Outcomes			
Any hemorrhagic transformation	167 (30.8)	33 (55.9)	<0.001
Parenchymal hematoma	70 (13.0)	16 (27.1)	0.006
Parenchymal hematoma, type 2	38 (7.0)	7 (11.9)	0.191
mRS at 3 months	3 (1–4)	5 (4–6)	<0.001
Good functional outcome (mRS 0–2)	264 (48.7)	7 (11.9)	<0.001
Mortality at 3 months	57 (10.5)	29 (49.2)	<0.001

Values are presented as *n* (%), mean ± standard deviation, or median (interquartile range). TIA, transient ischemic attack; CTA, computed tomography angiography; EVT, endovascular treatment; eGFR, estimated glomerular filtration rate; NIHSS, National Institutes of Health Stroke Scale; ASPECTS, Alberta stroke program early CT score; tPA, tissue plasminogen activator; mTICI, modified Thrombolysis in Cerebral Infarction; mRS, modified Rankin Scale. * *p*-value for trend. † ASPECTS was properly measured in 492 patients (92.8% of anterior circulation occlusion).

**Table 2 jcm-09-01471-t002:** Multivariate analysis of factors associated with acute kidney injury.

	Odds Ratio (95% Confidence Interval)	*p*-Value
Age, years	0.990 (0.961–1.022)	0.541
Sex, men	0.581 (0.316–1.057)	0.077
Hypertension	1.974 (0.978–4.201)	0.066
Diabetes mellitus	2.341 (1.283–4.269)	0.005
Statin medication prior to admission	1.734 (0.654–4.211)	0.242
Baseline renal function		
eGFR ≥90 mL/min/1.73 m^2^	Ref	
eGFR 60–89 mL/min/1.73 m^2^	0.616 (0.270–1.424)	0.249
eGFR 30–59 mL/min/1.73 m^2^	0.521 (0.211–1.309)	0.160
eGFR <30 mL/min/1.73 m^2^	1.434 (0.432–4.697)	0.551
CTA before EVT	2.112 (0.786–7.406)	0.181
Contrast dose, per 10 mL increase	1.107 (1.032–1.187)	0.004
NIHSS score on admission	1.041 (0.993–1.092)	0.095
Unsuccessful reperfusion	1.909 (1.019–3.520)	0.040

eGFR, estimated glomerular filtration rate; EVT, endovascular treatment; CTA, computed tomography angiography; NIHSS, National Institutes of Health Stroke Scale.

**Table 3 jcm-09-01471-t003:** Multivariate analysis of factors associated with a poor functional outcome at 3 months.

	Model 1		Model 2	
	Odds Ratio (95% Confidence Interval)	*p*-Value	Odds Ratio (95% Confidence Interval)	*p*-Value
Age, years	1.047 (1.025–1.071)	<0.001	1.047 (1.025–1.071)	<0.001
Sex, men	0.755 (0.501–1.134)	0.176	0.755 (0.501–1.135)	0.177
Hypertension	1.178 (0.769–1.805)	0.450	1.187 (0.775–1.820)	0.430
Diabetes mellitus	1.465 (0.934–2.309)	0.098	1.460 (0.929–2.303)	0.102
Baseline renal function				
eGFR ≥90 mL/min/1.73 m^2^	Ref		Ref	
eGFR 60–89 mL/min/1.73 m^2^	1.260 (0.678–2.354)	0.465	1.245 (0.670–2.324)	0.490
eGFR 30–59 mL/min/1.73 m^2^	0.914 (0.456–1.824)	0.798	0.894 (0.446–1.784)	0.750
eGFR <30 mL/min/1.73 m^2^	1.145 (0.396–3.423)	0.805	1.117 (0.385–3.346)	0.840
Presence of acute kidney injury	5.145 (2.177–13.850)	<0.001		
Stage of acute kidney injury				
No acute kidney injury			Ref	
Stage 1			2.938 (0.888–11.699)	0.094
Stage 2			13.709 (2.108–280.187)	0.022
Stage 3			6.028 (1.452–42.593)	0.030
Contrast dose, per 10 mL increase	1.080 (1.013–1.155)	0.021	1.078 (1.011–1.153)	0.025
White blood cell count	1.076 (1.011–1.148)	0.024	1.076 (1.011–1.149)	0.024
NIHSS score on admission	1.129 (1.089–1.174)	<0.001	1.130 (1.089–1.174)	<0.001
Onset to puncture time, min	1.001 (1.0001–1.002)	0.034	1.001 (1.0001–1.002)	0.029
Procedure time, min	1.012 (1.006–1.018)	<0.001	1.012 (1.006–1.018)	<0.001
Unsuccessful reperfusion	2.686 (1.640–4.468)	<0.001	2.672 (1.630–4.445)	<0.001
Parenchymal hematoma, type 2	4.438 (1.792–12.877)	0.003	4.510 (1.818–13.098)	0.002

Univariate analysis of factors associated with poor functional outcome is described in the Appendix A. eGFR, Estimated glomerular filtration rate; NIHSS, National Institutes of Health Stroke Scale.

**Table 4 jcm-09-01471-t004:** Multivariate analysis of factors associated with mortality at 3 months.

	Model 1		Model 2	
	Odds Ratio (95% Confidence Interval)	*p*-Value	Odds Ratio (95% Confidence Interval)	*p*-Value
Age, years	1.006 (0.978–1.036)	0.675	1.006 (0.978–1.037)	0.667
Sex, men	1.377 (0.785–2.450)	0.269	1.300 (0.733–2.330)	0.372
Diabetes mellitus	1.212 (0.663–2.168)	0.524	1.247 (0.676–2.252)	0.471
Statin medication prior to admission	1.776 (0.624–4.809)	0.267	1.705 (0.580–4.751)	0.317
Baseline renal function				
eGFR ≥90 mL/min/1.73 m^2^	Ref		Ref	
eGFR 60–89 mL/min/1.73 m^2^	2.515 (0.988–6.916)	0.062	2.438 (0.936–6.868)	0.078
eGFR 30–59 mL/min/1.73 m^2^	4.264 (1.598–12.361)	0.005	4.012 (1.465–11.935)	0.009
eGFR <30 mL/min/1.73 m^2^	3.949 (1.048–15.280)	0.043	3.609 (0.904–14.548)	0.068
Presence of acute kidney injury	8.164 (4.046–16.709)	<0.001		
Stage of acute kidney injury				
No acute kidney injury			Ref	
Stage 1			2.355 (0.660–7.265)	0.155
Stage 2			20.845 (5.907–82.054)	<0.001
Stage 3			13.670 (4.740–41.925)	<0.001
Contrast dose, per 10 mL increase	1.039 (0.958–1.125)	0.345	1.027 (0.944–1.113)	0.530
White blood cells	0.968 (0.891–1.047)	0.433	0.969 (0.889–1.050)	0.452
Platelets	1.008 (1.004–1.013)	<0.001	1.008 (1.004–1.012)	<0.001
NIHSS score on admission	1.115 (1.063–1.172)	<0.001	1.116 (1.063–1.173)	<0.001
Procedure time, min	1.004 (0.998–1.010)	0.193	1.005 (0.999–1.011)	0.085
Unsuccessful reperfusion	2.383 (1.294–4.377)	0.005	2.475 (1.332–4.586)	0.004
Parenchymal hematoma, type 2	5.176 (2.450–10.836)	<0.001	5.212 (2.422–11.074)	<0.001

Univariate analysis of factors associated with mortality is described in the Appendix A. eGFR, estimated glomerular filtration rate; NIHSS, National Institutes of Health Stroke Scale.

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
