# Peer review of "Acute Kidney Injury after Endovascular Treatment in Patients with Acute Ischemic Stroke"

_jcm, 2020, doi:10.3390/jcm9051471_

Round 1

Reviewer 1 Report

General comment:

In this paper, authors investigated the frequency of AKI after acute ischemic stroke endovascular treatment, factors related to its incidence and if AKI could influence clinical outcomes and mortality.

The study is well conducted and described, and results are quite interesting, in line with the recent literature.

The main concerns are listed in the limitation section, and the authors are fully aware of them.

Still, the main flaw of the study remains the lack of a correct definition of the amount of contrast that is used during the procedures. The estimation of contrast medium used is only an approximation made by assigning a specific amount of contrast to each image series and routine procedural dose. Additionally, one of main findings of this study is related the amount of contrast used, that is only estimated.

The authors should find an implementation of how contrast medium was estimated.

The authors should also add a section to specify the EVT manoeuvres (direct aspiration devices, stent  retriever), and when adjuvant procedures (Balloon guide  catheters, intracranial or extracranial angioplasty, and/or stenting) are needed. These variables should be taken into account for a correlation to AKI.

Specific Comments:

- Please to rephrase the sentence line 145

- Please rephrase the sentence line 225-229

-Please add Reference line 258

Author Response

Answers to comments

MS ID#: jcm-785336

MS TITLE: Acute kidney injury after endovascular treatment in patients with acute ischemic stroke

#1 reviewer :

We wish to thank the reviewer for his/her helpful review of our manuscript.

General comment:

In this paper, authors investigated the frequency of AKI after acute ischemic stroke endovascular treatment, factors related to its incidence and if AKI could influence clinical outcomes and mortality.

The study is well conducted and described, and results are quite interesting, in line with the recent literature.

The main concerns are listed in the limitation section, and the authors are fully aware of them.

Still, the main flaw of the study remains the lack of a correct definition of the amount of contrast that is used during the procedures. The estimation of contrast medium used is only an approximation made by assigning a specific amount of contrast to each image series and routine procedural dose. Additionally, one of main findings of this study is related the amount of contrast used, that is only estimated.

The authors should find an implementation of how contrast medium was estimated.

Answer) We fully agree with the reviewer's statement that the estimation of contrast medium used may be inaccurate. We also took great pain to calculate the exact amount of contrast used. In the previously published studies, recorded amount in the document was mostly used. We identified that this is not consistent with what was used in real practices. We thought most of them are overestimate, because some portion after dilution during the procedure are discarded without being used. The contrast agent remaining in the syringe or the contrast agent prep for use was not used in the actual procedure, but it was often recorded as a contrast medium used. To compensate for this, we did not use the recorded data, but recalculated according to the cerebral angiography images and procedure. The amount of contrast medium used in initial selection process was additionally assessed by querying each practitioner. We thought our method is as close to the actual usage as possible rather than a simple record. However, our method does not necessarily match the actual usage and can be underestimate. We have written this in the limitation section. In additional, we have attached a table for the approximate dose of contrast medium used for each procedure.

Table R. Reference dose of contrast used for each procedure during endovascular treatment

Working name

Dose (mL)*

Initial catheterization and selection of target artery

12-16

Roadmap image

4-6

Direct aspiration

8

Stent retriever

10-13

Microselection (including roadmap)

1

3D image of anterior circulation

16

3D image of posterior circulation

12

Final image

4-8

* Doses are approximate, we adjusted contrast dose estimation on each image. We recalculated the contrast dose in case of using diluted contrast agent.

Including all processes before and after retrieval of stent

The authors should also add a section to specify the EVT manoeuvres (direct aspiration devices, stent retriever), and when adjuvant procedures (Balloon guide catheters, intracranial or extracranial angioplasty, and/or stenting) are needed. These variables should be taken into account for a correlation to AKI.

Answer) Thank you for your comments and suggestions. AKI was developed in 22 of 289 patients (7.6%) using direct aspiration devices as a first-line device, and 37 of 312 patients (11.9%) in other groups. Although AKI rate was slightly lower in the direct aspiration group, there was no statistical significance (p = 0.107). The rate of adjuvant procedures such as balloon angioplasty, intracranial or extracranial stenting, tirofiban infusion was similar between two groups (26.6% in AKI and 25.4% in without AKI group, p = 0.973). There were no significant differences in univariate analysis, so the variables were not added to multivariate analysis.

Specific Comments:

- Please to rephrase the sentence line 145

Answer) We changed the sentence line 145.

“Patients with successful reperfusion used smaller amounts of the contrast agent than those with unsuccessful reperfusion (68.5±36.0 vs. 79.1±39.5 mL, p = 0.004).”

- Please rephrase the sentence line 225-229

Answer) Thank you for your suggestion. We rephrased the sentence line 225-229.

“The rate in our study is also higher than other recently reported study using EVT patients. This is probably because the patients enrolled in this study are older, with those with worse ASPECTS, or have less successful reperfusion rate. However, the rate is lower than the 20.9% of stroke patients who were admitted to the neurology intensive care unit.”

-Please add Reference line 258

Answer) Thank you for your advice. We added references.

“In patients with poor reperfusion, the possibility of complications such as brain edema or pneumonia increases [1,2].”

Again, thanks for your helpful comments and suggestions for the strengthening of this manuscript.

References

  1. Finlayson, O.; Kapral, M.; Hall, R.; Asllani, E.; Selchen, D.; Saposnik, G. Risk factors, inpatient care, and outcomes of pneumonia after ischemic stroke. Neurology 2011, 77, 1338-1345, doi:10.1212/WNL.0b013e31823152b1.
  2. Hassan, A.E.; Chaudhry, S.A.; Zacharatos, H.; Khatri, R.; Akbar, U.; Suri, M.F.; Qureshi, A.I. Increased rate of aspiration pneumonia and poor discharge outcome among acute ischemic stroke patients following intubation for endovascular treatment. Neurocrit. Care 2012, 16, 246-250, doi:10.1007/s12028-011-9638-0.

Reviewer 2 Report

This is a retrospective analysis from a  prospectively-maintained, multi-center registry of patients undergoing EVT. The investigators explored the incidence and impact of CA-AKI in the registry. Out of 601 patients, 59 (9.8%) developed AKI which was a predictor of poor functional outcome and mortality. 

The data are of high quality and the findings are relevant given the increasing number of EVT procedures performed.

I have the following suggestions for the investigators:

  • In the multivariable analysis for predictors of AKI (Table 2), the investigators adjusted for 10 variables when the outcome (AKI) occurred in only 59 patients. Ideally, the number of factors should be limited to 6 to avoid overfitting. 
  • How many patients received CT perfusion in addition to CTA? What is the average dose of contrast for CTA in participating centers?
  • 14% of the sample did not have a follow-up renal function and were excluded. As a sensitivity analysis, please calculate the incidence of AKI assuming these patients did not develop AKI. 
  • DM was overrepresented in the group that developed AKI. Did the investigators collect data on pre-admission medications like ACE inhibitors that may negatively impact kidney function? If not, this should be added to the limitations section. 
  • Unsuccessful reperfusion was also more common in the AKI group, although the procedural duration was essentially similar. What were the reasons for failed reperfusion which resulted in higher contrast use during EVT, e.g. ICAD, difficult access?
  • Pre-existing severe renal impairment was also higher in the AKI group. How was this defined?

Author Response

Answers to comments

MS ID#: jcm-785336

MS TITLE: Acute kidney injury after endovascular treatment in patients with acute ischemic stroke

#2 reviewer :

We wish to thank the reviewer for his/her helpful review of our manuscript.

This is a retrospective analysis from a prospectively-maintained, multi-center registry of patients undergoing EVT. The investigators explored the incidence and impact of CA-AKI in the registry. Out of 601 patients, 59 (9.8%) developed AKI which was a predictor of poor functional outcome and mortality.

The data are of high quality and the findings are relevant given the increasing number of EVT procedures performed.

I have the following suggestions for the investigators:

In the multivariable analysis for predictors of AKI (Table 2R), the investigators adjusted for 10 variables when the outcome (AKI) occurred in only 59 patients. Ideally, the number of factors should be limited to 6 to avoid overfitting.

Answer) The reviewer’s concern is pertinent. As the reviewer advised, it is correct to limit the number of factors to six in multivariate analysis. In order to solve the problem, we performed multivariate analysis again by stepwise regression. In the stepwise regression, we reduced the number of variables used by seven, and the results were not significantly different. Although there is one more variable used than your suggestion, these results suggested that our analysis is more acceptable. If the reviewer wants to substitute the Table 2, please let us know.

Table 2R. Multivariate analysis of factors associated with acute kidney injury

Odds ratio [95% confidence interval]

p-value

Sex, men

0.587 [0.328–1.039]

0.069

Hypertension

1.726 [0.901–3.479]

0.111

Diabetes mellitus

2.431 [1.345–4.397]

0.003

CTA before EVT

2.136 [0.845–7.415]

0.170

Contrast dose, per 10 mL increase

1.125 [1.053–1.202]

<0.001

NIHSS score on admission

1.044 [0.997–1.094]

0.069

Unsuccessful reperfusion

2.046 [1.116–3.693]

0.019

EVT, endovascular treatment, CTA, computed tomography angiography; NIHSS, National Institutes of Health Stroke Scale.

How many patients received CT perfusion in addition to CTA? What is the average dose of contrast for CTA in participating centers?

Answer) Thank you for your suggestions. Only 5 patients (0.7%) received CT perfusion in addition to CTA. Routine dose of contrast for CTA previous EVT was 90, 70, and 70-85 mL for each center. Two centers used a fixed dosage, one center adjusted contrast dose according to body weight. Average dose of contrast for CT previous EVT was 82.6±7.6 mL. We added the average contrast dose for CTA before EVT (line 143).

“CTA before EVT was performed in 510 patients (84.9%) and mean dose for CTA was 82.6±7.6 mL.”

14% of the sample did not have a follow-up renal function and were excluded. As a sensitivity analysis, please calculate the incidence of AKI assuming these patients did not develop AKI.

Answer) The reviewer’s comment is pertinent. Actually, 3 patients were excluded despite they did not develop AKI because initial eGFR was not available in them. So, strictly speaking, these patients are confirmed to have no AKI. The other excluded 95 patients had no clinical evidences of AKI, and blood tests performed after 1 week did not show any suspected AKI. However, by definition of AKI, they were excluded. In fact, it would be correct to analyze these patients as having no AKI, but they were excluded for the rigor of the study. If all 98 excluded patients said they had no AKI, the percentage of AKI decreased to 8.4%.

DM was overrepresented in the group that developed AKI. Did the investigators collect data on pre-admission medications like ACE inhibitors that may negatively impact kidney function? If not, this should be added to the limitations section.

Answer) Thank you for a good point. In fact, in addition to DM, initial glucose level showed a significant difference between patients with and without AKI, but DM was included in the multivariate analysis due to avoid multicollinearity. Unfortunately, we did not investigate ACE inhibitors when examining pre-admission medications. We will be able to do further investigations on the medication when conducting follow-up studies.

Unsuccessful reperfusion was also more common in the AKI group, although the procedural duration was essentially similar. What were the reasons for failed reperfusion which resulted in higher contrast use during EVT, e.g. ICAD, difficult access?

Answer) The reviewer’s question is pertinent. Mean number of endovascular techniques in the EVT process was higher in the failed reperfusion group (1.95±0.07 vs. 1.72±0.40, p = 0.006). The ratio of ICAD was similar between two groups (16.1% in unsuccessful reperfusion and 19.0% in successful reperfusion, p = 0.424). We thought difficult access is also one of the reasons for the difference in the amount of contrast medium used as you mentioned. I think these are the reasons for the large amount of contrast medium used in unsuccessful reperfusion.

Pre-existing severe renal impairment was also higher in the AKI group. How was this defined?

Answer) Good point. All included patients were bled prior to imaging to confirm baseline renal function. Patients identified as less than 30 mL/min/1.73m2 by calculating eGFR (CKD stage IV and V) were considered to have severe renal impairment. However, these contents overlap with the contents displayed directly above the table, so we will delete them from Table 1.

Again, thanks for your helpful comments and suggestions for the strengthening of this manuscript.

Round 2

Reviewer 1 Report

The Authors revised the manuscript. 

The instances reported in the previous revision were addressed.

I appreciated the clarification of contrast dose estimation, which ensures a more valuable methods accuracy.

Reviewer 2 Report

Thanks to the authors for addressing all my comments adequately. I do not have any other comments.